# Placental Infection Associated with SARS-CoV-2 Wildtype Variant and Variants of Concern

**DOI:** 10.3390/v15091918

**Published:** 2023-09-13

**Authors:** Ana Medel-Martinez, Cristina Paules, María Peran, Pilar Calvo, Sara Ruiz-Martinez, María Ormazabal Cundin, Alberto Cebollada-Solanas, Mark Strunk, Jon Schoorlemmer, Daniel Oros, Marta Fabre

**Affiliations:** 1Instituto de Investigación Sanitario de Aragón (IIS Aragon), 50009 Zaragoza, Spain; amedel@iisaragon.es (A.M.-M.); cristinapaules@hotmail.com (C.P.); mariaperanfernandez@hotmail.com (M.P.); pilarcalvocarod89@gmail.com (P.C.); sruizmart@gmail.com (S.R.-M.); danoros@gmail.com (D.O.); martafabreestremera@gmail.com (M.F.); 2Placental Pathophysiology & Fetal Programming Research Group, B46_20R & GIIS-028 del IISA, 50009 Zaragoza, Spain; 3Obstetrics Department, Hospital Clínico Universitario Lozano Blesa, 50009 Zaragoza, Spain; 4Red RICORS “Primary Care Interventions to Prevent Maternal and Child Chronic Diseases of Perinatal and Developmental Origin”, RD21/0012/0001, Instituto de Salud Carlos III, 28029 Madrid, Spain; 5Biochemistry Department, Hospital Clínico Universitario Lozano Blesa, 50009 Zaragoza, Spain; 6Instituto Aragonés de Ciencias de la Salud (IACS), Centro de Investigación Biomédica de Aragón (CIBA), 50009 Zaragoza, Spain; mcormazabal.iacs@aragon.es (M.O.C.); acebolladaso.iacs@aragon.es (A.C.-S.); mhpstrunk.iacs@aragon.es (M.S.); 7ARAID Foundation, 50009 Zaragoza, Spain

**Keywords:** SARS-CoV-2, SARS-CoV-2 variants of concern, pregnancy, waves of COVID-19

## Abstract

The original SARS-CoV-2 lineages have been replaced by successive variants of concern (VOCs) over time. The aim of this study was to perform an assessment of the placental infection by SARS-CoV-2 according to the predominant variant at the moment of COVID-19 diagnosis. This was a prospective study of SARS-CoV-2-positive pregnant women between March 2020 and March 2022. The population was divided into pregnancies affected by COVID-19 disease during 2020 (Pre-VOC group) and pregnancies affected after December 2020 by SARS-CoV-2 variants of concern (VOC group). The presence of virus was assessed by RT-PCR, and the viral variant was determined by whole genome sequencing. A total of 104 placentas were examined, among which 54 cases belonged to the Pre-VOC group and 50 cases belonged to the VOC group. Sixteen positive placental RT-PCR tests for SARS-CoV-2 were reported. The NGS analysis confirmed the SARS-CoV-2 lineage in placenta tissue. All samples corresponded to the Pre-VOC group, whereas no placental presence of SARS-CoV-2 was detected in the VOC group (16, 29.6% vs. 0, 0.0% *p* = 0.000). Preterm birth (9, 16.7% vs. 2, 4%; *p* = 0.036) and hypertensive disorders of pregnancy (14, 25.9% vs. 3, 6%; *p* = 0.003) were more frequent in the Pre-VOC group than in the VOC group. Finally, the VOC group was composed of 23 unvaccinated and 27 vaccinated pregnant women; no differences were observed in the sub-analysis focused on vaccination status. In summary, SARS-CoV-2-positive placentas were observed only in pregnancies infected by SARS-CoV-2 wildtype. Thus, placental SARS-CoV-2 presence could be influenced by SARS-CoV-2 variants, infection timing, or vaccination status. According to our data, the current risk of SARS-CoV-2 placental infection after maternal COVID disease during pregnancy should be updated.

## 1. Introduction

Since the coronavirus disease (COVID-19) outbreak caused by severe acute respiratory syndrome coronavirus 2 (SARS-CoV-2), the virus has been constantly changing [1,2,3]. The initial period of the pandemic up to December 2020, when SARS-CoV-2 wildtype was the predominant variant, has been referred to as pre-variants of concern (Pre-VOC) era. Then, the original SARS-CoV-2 lineages have been replaced by successive variants of concern (VOCs) (Alpha, Beta, Gamma, Delta, Omicron) over time [4]. These variants exhibit higher transmissibility and cause COVID-19 disease of lower severity (with associated lower mortality rates) compared to SARS-CoV-2 wildtype [5,6].

Apart from this evolution, since May 2021, the COVID-19 vaccines have become available for pregnant women, and their effectiveness has been demonstrated in the pregnant population [7,8,9]. 

The pregnancy complications associated with SARS-CoV-2 infection have become an issue of concern. Reports have shown an increase in maternal and neonatal morbidity and mortality in pregnant women with COVID-19 diagnosis during the first months of the pandemic [10,11]. On the other hand, the literature is undecided about the relationship between VOCs, vaccination and maternal outcomes [7,12,13,14]. 

In the context of maternal disease severity or vertical transmission to the fetus or newborn, several studies have studied the presence of SARS-CoV-2 virus in placenta. Whereas some studies have detected SARS-CoV-2 in 95–100% of analyzed placentas [15,16], others have only detected around 6–20% [17,18,19] and some cannot detect SARS-CoV-2 at all in placentas [20,21]. This variability can be explained, because different studies describe pregnant women with different grades of COVID-19 severity, and different methodologies have been used for viral RNA detection. In addition, variability might depend on the viral linage and on the vaccination status of women suffering COVID-19 disease. Although the SARS-CoV-2 lineages have changed in the course of the pandemic, the relationship between the predominant variant causing COVID-19 disease and either placental infection or the lineage actually infecting the placenta has hardly been reported on.

The aim of this study was to analyze the frequency of placental infection by SARS-CoV-2 in SARS-CoV-2-positive women during pregnancy. We analyzed samples of SARS-CoV-2-positive women during 2020, when the Pre-VOC predominated, and samples collected during waves of VOC (Appendix A).

## 2. Materials and Methods

We performed a prospective study in a Spanish tertiary care hospital (Hospital Clínico Lozano Blesa, Zaragoza, Spain) between March 2020 and March 2022. The inclusion criteria were SARS-CoV-2 infection during the pregnancy and placenta tissue available for analysis. The study was divided into two periods according to the date of their positive RT-PCR test and the periods of dominance of the SARS-CoV-2 variants. The Pre-VOC group was defined as women who tested positive for SARS-CoV-2 during 2020 (when the wildtype strain was predominant). The VOC group was defined as women who tested positive during 2021 and 2022 (era of all the other variants).

SARS-CoV-2 infection was diagnosed based on the positive RT-PCR test for SARS-CoV-2 from nasopharyngeal swabs. RT-PCR test kits from different companies were used: Viasure (CerTest Biotec, Zaragoza, Aragón, Spain), M2000 SARS-CoV-2 Assay (Abbott RealTime SARS-CoV-2 Assay, Abbott Molecular, Abbott Park, Green Oaks, IL, USA), TaqPath COVID-19 (Thermo Fisher Scientific, Waltham, MA, USA) and Alinity SARS-CoV-2 (Abbott Alinity, Abbott Molecular, Abbott Park, Green Oaks, IL, USA). As suggested by the manufacturer for nasopharyngeal specimens, cycle threshold (CT) values below 37 were taken as positive. Symptoms of COVID-19 have been divided into 3 types [22]: asymptomatic infection refers no clinical symptoms or signs; mild infection refers to symptoms such as fever, cough, headache, anosmia and asthenia; and severe infection refers dyspnea and hypoxemia accompanied by chest imaging compatible with pneumonia and respiratory infection. To determine the percentage of predominant variants of SARS-CoV-2 circulating in our patient recruitment area during sample collection, data were downloaded from the official local and national public sequence database [23] (Appendix A).

The collection of placental tissue samples, posterior treatment and storage in RNAtmLater, and subsequent purification of RNA and SARS-CoV-2 RT-PCR analysis was carried out as described in a previous study [24]. The RNA from placentas with a positive in SARS-CoV-2 RT-PCR was analyzed by NGS to identify SARS-CoV-2 lineage, using a modified version of the COVID-19 ARTIC v4 Illumina library construction and sequencing protocol v4. Details on sequencing procedures and subsequent data analysis are provided in the Appendix B. Typical coverage of the SARS-CoV-2 genome obtained by whole genome sequencing is shown in Appendix A.

Small for gestational age (SGA) was defined as birthweight below the 10th centile according to local standards [25]. Hypertensive disorders of pregnancy (HDPs) were divided into 4 categories, which were defined according to the criteria proposed by International Society for the Study of Hypertension in Pregnancy (ISSHP) [26]: preeclampsia (PE), chronic hypertension, chronic hypertension with superimposed PE and gestational hypertension.

Clinical characteristics, laboratory results, and maternal and neonatal outcomes were collected from medical records. All patients provided written informed consent. The study was approved by the Research Ethics Committee of the Community of Aragon (C.I. PI21/155 and COL21/000), and all patients provided written informed consent.

Statistical analysis was performed using SPSS 22.0. Categorical variables were presented as frequencies or percentages. Continuous variables were presented using mean ± standard deviation (SD), median, or range. For continuous variables, Shapiro–Wilk tests of normality were used to evaluate the distributions. Data were analyzed using Student’s *t* test or Mann–Whitney U test for continuous variables and the Pearson χ2 test for the categorical ones. Statistical significance was considered *p* < 0.05.

## 3. Results

Fifty-four women in the Pre-VOC group and fifty women in the VOC group were included. Demographic and clinical data, as well as a description of SARS-CoV-2, are shown in Table 1. The study group was comparable regarding maternal age, maternal weight, Caucasian ethnicity and primiparous. There were no significant differences for gestation age at birth, birth weight, SGA, umbilical arteria pH < 7.1, induced labor and caesarean. Nevertheless, preterm birth and HDP were more frequent in the Pre-VOC group than in the VOC group (*p* = 0.036, and *p* = 0.003, respectively).

Description of SARS-CoV-2 infection is presented in Table 1. No significant differences between trimester of SARS-CoV-2 maternal infection among cohorts were found (*p* = 0.083). Maternal diagnosis of COVID-19 was most frequent during the third trimester, 67 mothers (64.4%), whereas 33 women (31.7%) were in the second trimester and only four women (3.8%) were in the first trimester. However, we found significant differences according to the timing of maternal SARS-CoV-2 infection (*p* = 0.001) and the interval average from diagnosis of SARS-CoV-2 to delivery (Pre-VOC: 75.0 ± 54.2 days vs. VOC group: 50.3 ± 57.8, *p* = 0.027). Although no significant differences were found in the overall severity of SARS-CoV-2 symptoms between the groups (*p* = 0.184), the vast majority of patients with severe symptoms, four of five, were found in the Pre-VOC group.

We analyzed placental tissue from 104 patients by RT-PCR. Sixteen placentas were positive to SARS-CoV-2, and all were included in the Pre-VOC group. To confirm the presence and lineage of SARS-CoV-2 in the 16 positive placental samples identified, we performed whole viral genome sequencing by NGS. The assigned lineage in analyzed samples by whole genome sequencing corresponded to the predominant variant in our recruitment area at the time of infection of our study subjects. These results are shown in Table 2.

Additionally, we examined the vaccination status of study subjects. In the Pre-VOC group period, vaccination was still not available. In the VOC group, 27 women were vaccinated, whereas 23 women were unvaccinated (Figure 1). About vaccinated women, they were given viral vector (ChAdOx1-S (AstraZeneca)) or mRNA vaccines (mRNA-1273 (Moderna) or BNT162b2 (Pfizer-BioNTech)). Six women had only a single dose, whereas 20 women were completely vaccinated, and one had a booster vaccination. The average time between the last dose received and the COVID-19-positive result was 129.2 ± 85.1 days (Appendix A). Ten women had received the first dose before pregnancy, five women received the first dose during the first trimester, nine received the first dose during the second trimester and three received the first dose during the third trimester. Finally, we performed a sub-analysis to compare the Pre-VOC group with either the unvaccinated or the vaccinated VOC group (Table 3). Despite that in our sample we did not find differences among the overall maternal COVID-19 symptoms (*p* = 0.218), none of the cases or no severe cases occurred in the vaccinated VOC group.

## 4. Discussion

Our findings suggest that the frequency of SARS-CoV-2-positive placental tests may differ based on SARS-CoV-2 variants. We found a significantly higher frequency of positive placental RT-PCR SARS-CoV-2 in the placentas from women infected with SARS-CoV-2 wildtype compared with women infected with variants of concern of SARS-CoV-2, even adjusting for the vaccination status of the mother.

There are very few studies published describing the relation between SARS-CoV-2 variants of concern and placental health. Shanes et al. [27] detailed that maternal vascular malperfusion is a feature of SARS-CoV-2 infection during pregnancy and proposed that the lesion frequency changed with the predominant circulating variant. These findings are consistent with our results, given that positive placental SARS-CoV-2 could vary depending on the SARS-CoV-2 variants.

In fact, only a very limited number of studies combine data about the relationship between SARS-CoV-2 variants and positive placental SARS-CoV-2 tests. Wierz et al. [28], reporting on an isolated case, detected the presence of the Alpha variant in placental tissue using MALDI-TOF technology. Argueta et al. [29] described a cohort of placental samples from mothers positive for SARS-CoV-2 at delivery, where forty-two percent had detectable RNA, and underscored two placentas from mothers infected with the Alpha variant. A single case of SARS-CoV-2 variant Delta in a placenta after two successive COVID-19 episodes in unvaccinated woman in the same pregnancy has also been reported [30]. These isolated cases carried SARS-CoV-2 variants of concern in placenta tissue in contrast to our cohort, in which we only detected the SARS-CoV-2 wildtype. This observation could be explained by the frequency of SARS-CoV-2 infection early during pregnancy, vaccination status or herd immunity. Independent of these variables, in our limited analyses, no placental infection of SARS-CoV-2 was detected starting in 2021 (Table 1). Altogether, very few cases of placental presence of SARS-CoV-2 VOC have been described, suggesting it is rare. However, whether the frequency of placental infection, replication and/or symptoms in placenta are different between Pre-VOC on the one hand, and VOC and posterior variants on the other, remains to be established. Altered angiogenesis and enhanced vascular alterations are associated with both COVID-19 and PE [31]. We hypothesize that the resulting increased permeability of the blood vessels may allow more infiltrations of SARS-CoV-2 infected cells into the placenta and subsequent placental infection. A recent study provides evidence that disease severity is reduced in pregnant women infected by newer variants such as the omicron strain compared to women infected with wildtype SARS-CoV-2 at the beginning of the pandemic [32]. Our data suggest that the presence of placental SARS-CoV-2 is less frequent in mothers infected with VOC compared to SARS-CoV-2 wildtype. Therefore, if placental infection was a function of disease severity, the reduced frequency we observe after 2020 could be explained by the more limited vascular damage associated with less severe disease.

Pregnant women and their fetuses represent a high-risk population during infectious disease outbreaks [33]. For several viral diseases, severity is dependent on the trimester of infection. In the case of Rubella [34], the risk of fetal infection is highest in the first trimester especially prior to 10 weeks of gestation. On the other hand, severe disease was more prevalent among women in the third trimester of pregnancy during an influenza pandemic (H1N1). Moreover, from implantation and trophoblast invasion during early pregnancy onwards, successful pregnancy requires an environment that is tolerant toward maternal/fetal immunological differences [35]. In light of these combined observations, we analyzed the frequency of placental SARS-CoV-2 as a function of the trimester of infection. We included women infected by SARS-CoV-2 during all stages or trimesters of pregnancy in contrast to many studies that focus on women with a positive SARS-CoV-2 diagnosis at labor on hospital admission [36,37]. The vast majority of SARS-CoV-2-positive placentas (12 out of 16) were delivered by women who were diagnosed with COVID-19 disease more than 10 days before delivery (Table 2). Indeed, the seven cases that correspond to women who were diagnosed with COVID-19 before the third trimester all belong to the Pre-VOC group. While we cannot exclude the possibility that these mothers were re-infected later during pregnancy, an equally plausible explication resides in persistent infection. This timeframe would define these cases as post-COVID-19 syndrome, according to most actual definitions [38]. Glynn et al. [39] studied placental pathology, differentiating between acute or nonacute SARS-CoV-2 based on infection <14 or ≥14 days from delivery admission. This study provides evidence that histologic lesions in the placenta may differ based on the timing of SARS-CoV-2 infection during pregnancy. Furthermore, a significantly lower placental weight was reported in the non-acute cohort, suggesting long-term sequelae in response to SARS-CoV-2 infection. As SARS-CoV-2 infection may be persistent after infection during early pregnancy, and distinct placental lesions may develop during this time, further research is called for to mechanistically understand the effects on both placental and fetal health and to evaluate potential effects on newborns.

Whereas it was not the primary outcome of this study, we showed worse maternal and perinatal outcome in the Pre-VOC group compared to the VOC group. In fact, the association with preeclampsia that was reported early in the pandemic [24,40] is confirmed by our data (Pre-VOC group 14/54 HDP cases vs. VOC group 3/50 HDP cases, *p* = 0.003). Several studies correlate a worse perinatal outcome such as preterm birth, small for gestational age or ICU admission with hypertensive disorder of pregnancy [41,42]. Consistent with these reports, we observed higher rates of preterm birth in the Pre-VOC group than in the VOC group (9, 16.7% vs. 2, 4%; *p* = 0.036). Our results support reported findings suggesting that successive waves of SARS-CoV-2 variants of concern are associated with decreased severity in pregnant women [31,43,44], while other studies show that the VOC cases of severe COVID-19 disease during pregnancy are limited to women that are unvaccinated [9,45,46,47]. We also addressed the potential influence of vaccination on the presence of SARS-CoV-2 in the placenta. We did not detect SARS-CoV-2 in placenta tissue from 27 vaccinated women. Our data do not show significant differences between vaccination status and COVID-19 complications. This was probably due to the small sample size as well as because the vast majority of vaccinated women contracted the virus when Omicron was the most dominant strain. All vaccinated women in our study received vaccines not adapted to BA.4 and BA.5 Omicron variants. However, several studies published support the vaccine effectiveness in the pregnant population, showing a reduced risk for severe symptoms and complications in vaccinated women [9,48,49].

We assume that the sample size of this study is insufficient to provide evidence about the clinical outcomes associated with SARS-CoV-2 variants of concern, but it was not the main objective of this study. We consider that the prospective sampling of 106 placentas during the course of the pandemic provides enough consistency to evaluate the risk of placental infection over time as a function of different variants and clinical scenarios. Moreover, to collect the placentas of all SARS-CoV-2-positive pregnancies during the period of study, we needed to know at delivery time which women had passed COVID-19 during pregnancy and to have research staff available for sample collection and preservation. These requirements were very difficult to meet at certain times of the pandemic, and a much more limited number of samples was collected. This may have caused selection bias. It is important to note that the detection of the viral genome in the placental tissue does not necessarily mean that active infectious particles have been detected. However, the genome-wide coverage that we obtain in most positive samples (see Appendix A) would suggest that replication occurs and active infection may take place. Moreover, we now confirm that the lineage infecting the placenta corresponds to the predominant lineage at the moment of SARS-CoV-2 infection and not to rare variants with tropism for placenta.

In summary, SARS-CoV-2-positive placentas were observed only in pregnancies infected by the SARS-CoV-2 wildtype. The frequency of placental SARS-CoV-2 may be determined by the SARS-CoV-2 variant, infection timing, or vaccination status of the mother. According to our data, the current risk of SARS-CoV-2 placental infection after maternal COVID disease during pregnancy should be updated.

## Figures and Tables

**Figure 1 viruses-15-01918-f001:**
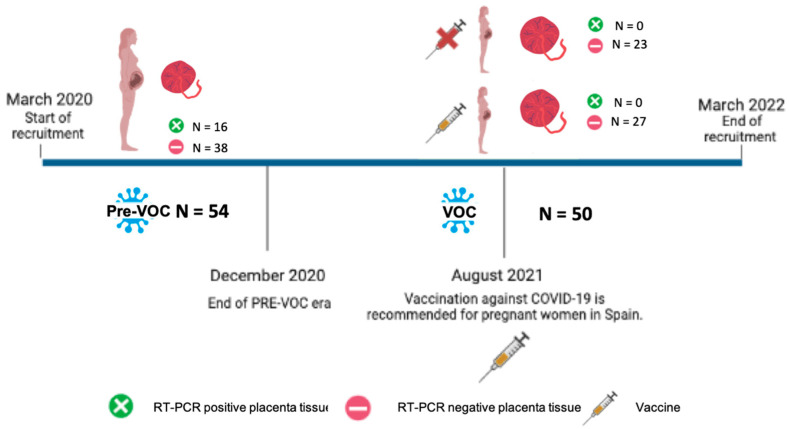
Analysis of association of SARS-CoV-2 placental infection with variant of concern era and vaccination. Pre-VOC group N = 54. VOC group = 50, including 23 unvaccinated women and 27 vaccinated women. Overall, 16 placentas were positive to SARS-CoV-2. All were included in the Pre-VOC group, no women were included in the VOC group.

**Table 1 viruses-15-01918-t001:** Demographic and obstetrical data and SARS-CoV-2 infection characteristics.

	Pre-VOC (*n* = 54)	VOC (*n* = 50)	*p*
Demographic maternal data			
Maternal age (SD), years	31.5 (5.9)	32.8 (5.4)	0.311
Maternal weight (SD), kg	69.5 (12.6)	67.9 (13.1)	0.330
Caucasian (%)	30 (55.5)	32 (64.0)	0.381
Primiparous (%)	24 (44.4)	18 (36.0)	0.381
Maternal and neonatal outcome at delivery			
Gestational age at birth mean (SD), days	272.2 (20.3)	274 (7.4)	0.365
Birth weight, mean (SD), grams	3114.5 (609.3)	3136.7 (343.6)	0.224
Small for gestational age neonate (%)	7 (13.0)	4 (8.0)	0.411
Preterm birth (%)	9 (16.7)	2 (4.0)	0.036
Hypertensive disorders of pregnancy (%)	14 (25.9)	3 (6.0)	0.003
Labor induction (%)	23 (42.6)	16 (52.0)	0.338
Cesarean delivery (%)	8 (14.8)	12 (24.0)	0.235
Umbilical artery pH < 7.10 (%)	2 (3.7)	2 (4.0)	0.984
Description of SARS-CoV-2 infection			
Trimester of SARS-CoV-2 infection			0.083
1° (%)	4 (7.4)	0 (0.0)	
2° (%)	19 (35.2)	14(28)	
3° (%)	31 (57.4)	36 (72)	
Time between SARS-CoV-2 diagnosis and delivery, days (%)			0.001
<10 days	7 (13.0)	23 (46.0)	
11–84 days	23 (42.6)	14 (28.0)	
>84 days	24 (44.4)	13 (26.0)	
Interval from diagnosis of SARS-CoV-2 to delivery (SD), days	75.0 (54.2)	50.3 (57.8)	0.027
COVID-19 Symptoms			0.184
No (%)	23 (42.6)	29 (58.0)	0.085
Mild (%)	27 (50.0)	20 (40.0)	0.204
Severe (%)	4 (7.4)	1 (2.0)	0.206
RT-PCR-positive placenta tissue (CT < 37), (%)	16 (29.6)	0 (0.0)	0.000

SD, standard deviation; CT, cycle threshold value RT-PCR.

**Table 2 viruses-15-01918-t002:** Description of sixteen positive placental SARS-CoV-2 cases. Assignment of SARS-CoV-2 lineage in each sample positive for placental SARS-CoV-2 based on the results of whole genome sequencing (see Appendix A for details).

	Diagnosis SARS-CoV-2			Delivery and Placenta Tissue Results						
N	GA at Diagnosis	PS	SYM	GA at Delivery	Interval SARS-CoV-2—Delivery (d)	RT-PCR Result Nasopharyngeal at Delivery Time	CT RT-PCR Nasopharyngeal	CT RT-PCR Placenta Tissue	Lineage	Correlation between PS and Placenta Tissue
1	38 + 3	Pre-VOC	S	38 + 4	1	Positive	33.5	15.1	B.1.177	Yes
2	25 + 1	Pre-VOC	M	39 + 2	99	Negative		33.9	B.1	Yes
3	31 + 4	Pre-VOC	A	41 + 0	74	Negative		30.2	B.1	Yes
4	24 + 6	Pre-VOC	M	37 + 1	86	Negative		31.7	B.1.177	Yes
5	25 + 6	Pre-VOC	M	40 + 1	100	Negative		33.1	-	-
6	32 + 0	Pre-VOC	A	40 + 1	85	Negative		31.1	B.1	Yes
7	21 + 3	Pre-VOC	A	36 + 5	107	N/A		33.4	B.1.177	Yes
8	24 + 4	Pre-VOC	M	39 + 5	106	Negative		32.4	B.1	Yes
9	36 + 2	Pre-VOC	A	38 + 0	12	Positive	34.8	35.2	B.1.177	Yes
10	30 + 0	Pre-VOC	A	40 + 2	72	Negative		24.5	B.1	Yes
11	37 + 6	Pre-VOC	A	38 + 1	2	Positive	35.3	33.9	-	-
12	17 + 5	Pre-VOC	M	38 + 2	151	Negative		32.0	B.1	Yes
13	27 + 5	Pre-VOC	M	37 + 0	68	Negative		23.3	B.1.177	Yes
14	32 + 3	Pre-VOC	A	41 + 2	64	Negative		32.5	B.1	Yes
15	8 + 0	Pre-VOC	A	35 + 2	191	Negative		28.7	B.1.177	Yes
16	41 + 0	Pre-VOC	A	41 + 1	8	Positive	35.5	33.7	-	-

GA, gestational age (week + days); PS, predominant strain at the moment of SARS-CoV-2 diagnosis; d, days; SYM, symptoms of SARS-CoV-2 infection; S, severe; M, mild; A, asymptomatic; -, absence of lineage assignment; CT, cycle threshold value RT-PCR; N/A, not available.

**Table 3 viruses-15-01918-t003:** Results of sub-analysis focused on vaccination status.

	Pre-VOC (*n* = 54)	VOC Unvaccinated (*n* = 23)	VOC Vaccinated (*n* = 27)	*p*
Hypertensive disorders of pregnancy, (%)	14 (25.9)	2 (8.7)	1 (3.7)	0.022
Preterm birth, (%)	9 (16.7)	1 (4.3)	1 (3.7)	0.110
Small for gestational age, (%)	7 (13.0)	1 (4.3)	3 (11.1)	0.528
Birth weight, mean (SD), g	3114.5 (609.3)	3168.3 (308.4)	3290.7 (367.2)	0.331
Trimester of SARS-CoV-2 infection				0.148
1° (%)	4 (7.4)	0 (0.0)	0 (0.0)	
2° (%)	19 (35.2)	4 (17.4)	10 (37)	
3° (%)	31 (57.4)	18 (78.3)	17 (63)	
Interval from diagnosis of SARS-CoV-2 to delivery (SD), d	75.0	43.9	55.8	0.07
COVID-19 Symptoms				0.370
No (%)	23 (42.6)	12 (52.2)	17 (63)	0.218
Mild (%)	27 (50)	10 (43.5)	10 (37)	0.534
Severe (%)	4(7.4)	1 (4.3)	0 (0.0)	0.338
COVID-19 symptomatic, (%)	31 (57.4)	11 (47.8)	10 (37)	0.218
RT-PCR-positive placenta tissue (CT < 37), (%)	16 (29.6)	0 (0.0)	0 (0.0)	0.000

SD, standard deviation; CT, cycle threshold value RT-PCR.

## Data Availability

All data are presented in this study. Original data are available upon request from the corresponding author.

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
