# Peer review of "Placental Infection Associated with SARS-CoV-2 Wildtype Variant and Variants of Concern"

_viruses, 2023, doi:10.3390/v15091918_

Round 1

Reviewer 1 Report

The manuscript is of high relevance to understanding SARS-CoV-2 infection in the placenta. The presentation of methodology and results is scientifically appropriate; however, the introduction and discussion parts need to be modified as follows;

Introduction part) Add a brief description of why the authors classified the participants with the VOC era to understand the context of the manuscript.

Discussion part) There was a significant difference in hypertensive disorders during pregnancy between pre-VOC and VOC groups. The difference in the VOCs and the SARS-CoV-2 infection in the placenta might contribute to preterm birth. The results of the hypertensive disorders during pregnancy should be included in the paragraph in Lines 303-319 discussion.

The authors addressed the post-COVID-19 syndrome for the time difference of the PCR positive between maternal diagnosis and delivery (Line 293). Would you add a brief description of the mechanistic background?

Minor

Graphical abstract (Line 70) may not be provided. Supplemental Figure 1 can be cited here.

Author Response

Dear Reviewer,

Re: “Placental infection associated with SARS-CoV-2 wildtype variant and variants of concern”.

Thank you for your useful comments. Please find enclosed a point-by-point.

Thank you very much for your kind attention, and I look forward to hearing from you.

Yours sincerely,

Marta Fabre, MD

Introduction part) Add a brief description of why the authors classified the participants with the VOC era to understand the context of the manuscript.

Thank you for this suggestion. SARS-CoV-2 has consistently mutated over the course of the pandemic, resulting in variants that are different from the original SARS-CoV-2. The major circulating variants (Alpha, Gamma, Delta, and Omicron) have changed considerably the epidemiology, presentation and transmissibility of COVID-19 disease. We wanted to investigate if these variants had different impact in the placenta tissue.

It has been included on line 43 “These variants exhibit higher transmissibility, and cause COVID-19 disease of lower severity (with associated lower mortality rates) compared to SARS-CoV-2 wildtype [5,6]”.

Discussion part) There was a significant difference in hypertensive disorders during pregnancy between pre-VOC and VOC groups. The difference in the VOCs and the SARS-CoV-2 infection in the placenta might contribute to preterm birth. The results of the hypertensive disorders during pregnancy should be included in the paragraph in Lines 303-319 discussion.

The following lines have been included on line 234 to address this point: “Several studies correlate a worse perinatal outcomes such as preterm birth, small for gestational age or ICU admission with HDP [39,40]. Consistent with these reports, we observed higher rates of preterm birth in the Pre-VOC group than in the VOC group (9, 16.7% vs 2, 4%; p=0.036).

The authors addressed the post-COVID-19 syndrome for the time difference of the PCR positive between maternal diagnosis and delivery (Line 293). Would you add a brief description of the mechanistic background?

We added this description on line 211: “Pregnant women and their fetuses represent a high-risk population during infectious disease outbreaks [33]. For several viral diseases, severity is dependent on the trimester of infection. In the case of Rubella [34], the risk of fetal infection is highest in the first trimester, especially prior to 10 weeks of gestation. On the other hand, severe disease was more prevalent among women in the third trimester of pregnancy during an influenza pandemic (H1N1). Moreover, from implantation and trophoblast invasion during early pregnancy onwards, successful pregnancy requires an environment that is tolerant to-wards maternal/fetal immunological differences [35]. In light of these combined observations, we analyzed the frequency of placental SARS-CoV-2 as a function of the trimester of infection”.

At the end of the paragraph, line 234, we added: As SARS-CoV-2 infection may be persistent after infection during early pregnancy, and distinct placental lesions may develop during this time, further research is called for to mechanistically understand the effects on both placental and fetal health, and to evaluate potential effects on newborn.

Graphical abstract (Line 70) may not be provided. Supplemental Figure 1 can be cited here.

We have moved the Graphical abstract to Supplemental Figure 1

Reviewer 2 Report

The work carried out by Medel Martinez et al. shows an exciting approach to SARS-CoV-2 infection during pregnancy, focusing on evaluating infection in placentas with different variants of this virus. Although only some of the works relate to this topic, the authors show interesting results and great importance for the scientific community. Clearly, its main contribution shows how the wild-type virus had a more infective behavior in the placentas obtained during the pandemic, unlike the placentas obtained from patients infected with the later variants. Additionally, the work shows a broad approach, considering the possibility of null infection and the prompt implementation of vaccination in pregnant women. However, the discussion requires more information that can reconsider why this phenomenon of the original virus is found in the placentas and why it does not occur with the other variants. Although immunity could be the explanation, not only because of the vaccines but also because of the natural infections that the population of women experienced, there is a vast literature concerning the phenomenon that the disease caused at the beginning of the pandemic in pregnant women, it is likely that some of this evidence can help deepen the discussion and can enrich the work that the authors are presenting.

The article has good writing and grammar, and I do not consider it to require a style correction.

Author Response

Dear Reviewer,

Re: “Placental infection associated with SARS-CoV-2 wildtype variant and variants of concern”.

Thank you for your useful comment.

You suggest we deepen the discussion as follows: However, the discussion requires more information that can reconsider why this phenomenon of the original virus is found in the placentas and why it does not occur with the other variants. Although immunity could be the explanation, not only because of the vaccines but also because of the natural infections that the population of women experienced, there is a vast literature concerning the phenomenon that the disease caused at the beginning of the pandemic in pregnant women, it is likely that some of this evidence can help deepen the discussion and can enrich the work that the authors are presenting

We somewhat altered the discussion and tried to address the issues mentioned. Added text starts on line 183, and some corrections were made to the preceding paragraph:

In fact, only a very limited number of studies combine data about the relationship between SARS-CoV-2 variants and positive placental SARS-CoV-2 tests. Wierz el at [28] reporting on an isolated case, detected the presence of the Alpha variant in placental tissue using MALDI-TOF technology. Argueta et al [29] described a cohort of placental samples from mothers positive for SARS-CoV-2 at delivery, where forty-two percent had detectable RNA, and underscored two placentas from mothers infected with the Alpha variant. A single case of SARS-CoV-2 variant Delta in a placenta after two successive COVID-19 episodes in unvaccinated woman in the same pregnancy has also been reported [30]. These isolated cases carried SARS-CoV-2 variants of concern in placenta tissue in contrast to our cohort, in which we only detected SARS-CoV-2 wildtype. This observation could be explained by the frequency of SARS-CoV-2 infection early during pregnancy, vaccination status or herd immunity. Independent of these variables, in our limited analyses, no placental infection of SARS-CoV-2 was detected starting in 2021 (Table 1). Altogether, very few cases of placental presence of SARS-CoV-2 VOC have been described, suggesting it is rare. However, whether the frequency of placental infection, replication and/or symptoms in placenta are different between preVOC on the one hand, and VOC and posterior variants on the other, remains to be established. Altered angiogenesis and enhanced vascular alterations are associated with both COVID-19 and PE [31]. We hypothesize that the resulting increased permeability of the blood vessels may allow more infiltrations of SARS-CoV-2 infected cells into the placenta, and subsequent placental infection. A recent study provides evidence that disease severity is reduced in pregnant women infected by newer variants such as the omicron strain, compared to women infected with wild type SARS-CoV-2 at the beginning of the pandemic[32]. Our data suggest that the presence of placental SARS-CoV-2 is less frequent in mothers infected with VOC compared to SARS-CoV-2 wildtype. Therefore, if placental infection were a function of disease severity, the reduced frequency we observe after 2020 could be explained by the more limited vascular damage associated with less severe disease.

.

Reviewer 3 Report

Thank you for the opportunity to read this manuscript. Please note comments and concerns that need to be revised. I believe the authors can improve the manuscript.

Abstract

The abstract summarizes the main result

Indicate percentage in these results: “Preterm birth (9 vs 2; p=0.036), and hyper- 29 tensive disorders of pregnancy (15 vs 3; p=0.003)”.

MAIN TEXT

Introduction: Please revise the objective presented in the last paragraph of this section. This is not clear.

Methods:

Describe the setting and locations of this study.

How was the sample size determined for the study?

Describe the strategy used of recruitment or to include cases in the study.

Results:

It is requested to present the number and % of excluded cases according to the main causes.

Table 3 is not clear. Is it a case description? I believe this table is not useful, please review.

Discussion:

It was not clear how the cases were recruited, which may cause selection bias. This aspect could be presented as a limitation.

Author Response

Dear Reviewer,

Re: “Placental infection associated with SARS-CoV-2 wildtype variant and variants of concern”.

Thank you for your positive comments on our manuscript. Please find enclosed a point-by-point description of changes incorporated into the manuscript.

Yours sincerely,

Marta Fabre, MD

The abstract summarizes the main result. Indicate percentage in these results: “Preterm birth (9 vs 2; p=0.036), and hypertenive disorders of pregnancy (14 vs 3; p=0.003)

Percentages have been added to the abstract.

Introduction: Please revise the objective presented in the last paragraph of this section. This is not clear.

Thank you for the advice. The text has been adapted and now reads on line 67 “The aim of this study was to analyze the frequency of placental infection by SARS-CoV-2 in SARS-CoV-2 positive women during pregnancy. We analyzed samples of SARS-CoV-2 positive women during 2020, when the pre-VOC predominated, and samples collected during waves of VOC (Supplemental Figure 1)”.

Methods: Describe the setting and locations of this study. How was the sample size determined for the study? Describe the strategy used of recruitment or to include cases in the study. It is requested to present the number and % of excluded cases according to the main causes.

Our original aim was to collect placentas of all SARS-CoV-2 positive pregnancies during the period of study. To this end, we needed to know which women had passed COVID-19 during the pregnancy at delivery time and to have research staff available for samples collection and preservation. These requirements at certain times of the pandemic were very difficult to meet, and a much more limited number of samples was collected. During the study period, 3804 women have given birth in our hospital, but we do not know the percentage of SARS-CoV-2 positive women to estimate the percentage of excluded cases.

We have removed the term “cohort”. The text has been modified on line 72 into “We performed a prospective study in a Spanish tertiary care hospital (Hospital Clínico Lozano Blesa, Zaragoza, Spain) between March 2020 and March 2022. The inclusion criteria were: SARS-CoV-2 infection during the pregnancy and placenta tissue available for analysis”.

Table 3 is not clear. Is it a case description? I believe this table is not useful, please review.

We have moved the table 3 to material supplementary.

It was not clear how the cases were recruited, which may cause selection bias. This aspect could be presented as a limitation.

A new limitation has been included on line 265 “To collect placentas of all SARS-CoV-2 positive pregnancies during the period of study, we needed to know at delivery time which women had passed COVID-19 during pregnancy and to have research staff available for sample collection and preservation. These requirements were very difficult to meet at certain times of the pandemic, and a much more limited number of samples was collected. This may have caused selection bias”.